# Emerging Roles of TRIM8 in Health and Disease

**DOI:** 10.3390/cells10030561

**Published:** 2021-03-05

**Authors:** Flaviana Marzano, Luisa Guerrini, Graziano Pesole, Elisabetta Sbisà, Apollonia Tullo

**Affiliations:** 1Institute of Biomembranes, Bioenergetics and Molecular Biotechnologies, National Research Council, CNR, 70126 Bari, Italy; f.marzano@ibiom.cnr.it (F.M.); g.pesole@ibiom.cnr.it (G.P.); 2Department of Biosciences, Università degli Studi di Milano, 20133 Milan, Italy; luisa.guerrini@unimi.it; 3Department of Biosciences, Biotechnology and Biopharmaceutics, University of Bari Aldo Moro, 70121 Bari, Italy; 4Institute of Biomedical Technologies, National Research Council, CNR, 70126 Bari, Italy; elisabetta.sbisa@ba.itb.cnr.it

**Keywords:** TRIM8, p53, NF-κB, JAK-STAT

## Abstract

The superfamily of TRIM (TRIpartite Motif-containing) proteins is one of the largest groups of E3 ubiquitin ligases. Among them, interest in TRIM8 has greatly increased in recent years. In this review, we analyze the regulation of TRIM8 gene expression and how it is involved in many cell reactions in response to different stimuli such as genotoxic stress and attacks by viruses or bacteria, playing a central role in the immune response and orchestrating various fundamental biological processes such as cell survival, carcinogenesis, autophagy, apoptosis, differentiation and inflammation. Moreover, we show how TRIM8 functions are not limited to ubiquitination, and contrasting data highlight its role either as an oncogene or as a tumor suppressor gene, acting as a “double-edged weapon”. This is linked to its involvement in the selective regulation of three pivotal cellular signaling pathways: the p53 tumor suppressor, NF-κB and JAK-STAT pathways. Lastly, we describe how TRIM8 dysfunctions are linked to inflammatory processes, autoimmune disorders, rare developmental and cardiovascular diseases, ischemia, intellectual disability and cancer.

## 1. Introduction

A delicate balance between protein synthesis and degradation is essential to maintain cell homeostasis. Deregulation of protein homeostasis in favor of either protein synthesis or protein degradation is detrimental, although in different ways. The role of the Ubiquitin–Proteasome System (UPS) is central in maintaining protein homeostasis, and alteration of the UPS has been linked to several pathological conditions.

Ubiquitination is one of the most prevalent post-translational protein modifications. It plays a key role in several cellular processes and physiological responses in inflammatory disorders, neurodegeneration, cancer, autoimmunity, infection and other human diseases. Ubiquitination can target proteins for degradation via the proteasome or selective autophagy, alter subcellular localization, affect activity and regulate interactions with other proteins.

The E3 Ub ligases function in association with an E1 ubiquitin-activating enzyme and an E2 ubiquitin-conjugating enzyme. There are only two human genes coding for E1 enzymes, 30–50 genes for E2 and over 600 E3 ligase genes, which constitute about 3% of human protein coding genes. These types of E3 ligases differ from each other depending on the type of catalytic domains: Really Interesting New Gene (RING), Homologous to E6-AP Carboxyl Terminus (HECT) or Ring-Between-Ring (RBR). The high number of E3 ligases is associated to their specificity in selectively targeting protein substrates [1,2].

Each ubiquitin molecule contains seven lysine residues: Lys6, Lys11, Lys27, Lys29, Lys33, Lys48 and Lys63; additional ubiquitins can be attached to each of these lysines to form chains of various lengths with different functions. Proteasome-dependent degradation is related to Lys48-linked chains while nonproteolytic roles of ubiquitin Lys63-linked chains function as signaling scaffolds for Nuclear Factor-κB (NF-κB) activity, DNA repair and intracellular trafficking [3,4,5]. Moreover, Lys63 polyubiquitin chains are involved in autophagy, recruiting several ubiquitin-binding proteins such as p62, Neighbor of BRCA1 gene 1 (NBR1) or Histone DeACetylase 6 (HDAC6), triggering the formation of inclusion bodies within the autophagic pathway. On the other hand, Lys6-linked ubiquitin chains function within a nondegradative route and have been linked to physiological roles in DNA damage repair. Proteasome signaling has been linked to Lys11, Lys27, Lys29 and Lys33 ubiquitin chains. Lys11 polyubiquitination increases at the end of mitosis, connecting polyubiquitination to cell cycle regulation. Lys27 ubiquitination has been observed in conjunction with mitochondrial damage.

In addition to the aforementioned seven internal lysine residues, N-terminal methionine ubiquitination has been identified by Iwai and colleagues as the action site for a novel RING E3 ligase complex capable of connecting ubiquitin molecules in a head-to-tail fashion [6,7]. Met1-linked linear chains are formed by the Linear Ubiquitin Assembly Complex (LUBAC): a 600-κD E3 ligase complex composed of two RBR ligases, the Heme-Oxidized Iron-responsive element-binding protein 2 ubiquitin Ligase-1L (HOIL-1L) and the HOIL-1L-interacting Protein (HOIP), in addition to SHank-Associated Rh domain-interacting ProteIN (SHARPIN). Linear ubiquitination is a new atypical nondegradative ubiquitin modification [1,2]. It has been shown that the RBR ligase domain of HOIL-1L is able to add Lys48 polyubiquitin chains to target proteins regardless of LUBAC.

The importance of the ubiquitin signaling system is highlighted by the fact that misregulation of ubiquitin signaling or functional impairment of the proteasome has been associated with several pathological conditions.

It is interesting to note that autosomal defects in LUBAC are associated with atypical autoinflammation and immunodeficiency.

Significant progress has been recently made regarding the discovery of different roles of polyubiquitination chains in different signaling pathways and how dysfunctions in these processes are involved in cancer, inflammatory disorders, autoimmunity, neurodegeneration, infection and other diseases.

Most tumors are prone to develop upon alterations in ubiquitination-mediated events. Furthermore, the analysis of the transcriptional profile of patients affected by lung adenocarcinoma and glioblastoma showed differential expression of HOIL-1L with higher levels being associated with decreased survival [1,2].

## 2. TRIM Family Proteins

The TRIM (TRIpartite Motif-containing)/RBCC family belongs to the RING family of ubiquitin E3 ligases, containing an N terminal RING domain, one or two B-Box motifs and the Coiled-Coil (CC) domain (RBCC) followed by a highly variable carboxyl-terminal domain, which allows the classification of TRIM proteins into 11 subgroups (Figure 1). In humans, TRIM proteins total more than 70 members, representing one of the largest groups of the E3 ligase RING family [8]. TRIM proteins are conserved throughout the metazoan kingdom. Their primary sequences show a relatively low similarity, barring a few members [9]. Only the cysteine and histidine that characterize the RING and B-box domains and the hydrophobic residues of the Coiled-Coil region are highly conserved because they are necessary to sustain the scaffold structure of the proteins. The intervening sequences between the domains of the TRIM proteins evolved rapidly to gain new physiological functions and specificity. The different domains of TRIM proteins are necessary to control the formation of higher order structures and cellular localization. Indeed, TRIM proteins are capable of associating in high molecular weight complexes through the interaction of Coiled-Coil domains. These complexes localize in specific cellular sub-compartments, such as cytoplasmic bodies or ribbon-like structures, which can be placed around the nucleus (TRIM13) or in the nucleus where they form “nuclear bodies” (TRIM8, 19, 30 and 32) or “nuclear sticks” (TRIM6). Some members such as TRIM24, 28 and 33 contain the bromo domain, which in the nucleus interacts with the acetylated lysines of histones [10]. Only few family members do not have a RING domain but are still considered TRIM/RBCC proteins since they retain all the other domains (B-boxes and Coiled-Coil) in the same order as the other members (Figure 1).

The proteins that belong to this family have the peculiarity of exerting a great variety of roles and different functions because of their ubiquitination or ubiquitin-like activity that, as reported above, not only tags the target proteins to be degraded at the proteasome level, but can stabilize or re-localize them in different cellular compartments by such modifications.

TRIM proteins are involved in the regulation of cellular homeostasis, cell cycle, senescence, apoptosis, differentiation, specific metabolic pathways, meiosis and protein quality control.

## 3. TRIM8, a Double-Edged Weapon

In recent years, there has been growing interest in TRIM8 protein research. The TRIM8 gene is located on the 10q24.3 chromosome and transcribes an mRNA of about 3.0 κb that is translated into a protein of 551 a.a. with a molecular weight of 61.5 κDa.

The protein structure consists of a RING finger domain at the N-terminal, two B-box domains, a Coiled-Coil domain and an RFL-like domain at the C-terminal (Figure 1) [8]. Moreover, TRIM8 protein contains a Nuclear Localization Signal (NLS), which allows translocation and functioning in the nucleus. The Coiled-Coil domain of TRIM8 permits the formation of Nuclear Bodies (NBs) similar to TRIM19/PML, regulating the activity of important cellular proteins through protein–protein interactions [10]. Recently, an important role in the mitotic spindle machinery has been described for TRIM8. From the analysis of the TRIM8 interactome in primary mouse embryonic neural stem cells, it was found that TRIM8 interacts with KIFC1 and KIF11/Eg5 kinesins, two master regulators of mitotic spindle assembly and cytoskeleton reorganization. In particular, during mitosis TRIM8 localizes at the mitotic spindle playing a role in centrosome separation. TRIM8 knock-down slows centrosome separation at the prometaphase resulting in chromosome instability as aneuploidic cells and micronuclei formation [11].

TRIM8 is involved in many cell reactions in response to different stimuli such as genotoxic stress and attacks by viruses or bacteria, playing a central role in the immune response and orchestrating various fundamental biological processes such as cell survival, innate immune response, carcinogenesis, autophagy, apoptosis, differentiation and inflammation. Its dysfunction is linked to cancer, inflammatory processes and autoimmune disorders. The involvement of TRIM8 in such a plethora of cellular functions is fundamentally linked to its involvement in the regulation of three pivotal cellular signaling pathways: the p53 tumor suppressor signaling pathway, the NF-κB pathway (Nuclear Factor kappa-light-chain-enhancer of activated B cells) and STAT3 (Signal Transducer and Activator of Transcription 3) of the JAK-STAT pathway. The TRIM8 liaison with these three pathways determines its dual role in cancer as oncogene or tumor suppressor functions [12].

## 4. Regulation of TRIM8 Gene Expression

TRIM8 is ubiquitously expressed in murine and human tissues with highest expression in the central nervous tissue, kidney and lens, and with lower expression in the gut as reported by in situ hybridization studies in mouse embryos [10]. Moreover, it is also expressed in undifferentiated embryonic stem cells, suggesting that TRIM8 could play an important role in maintaining pluripotency [13]. It seems that TRIM8 turnover is high and is stabilized following genotoxic stress [14]. In particular, under stress conditions, such as UV, p53 promoted the transcription of TRIM8 by binding to the p53 responsive element present in the first intron. TRIM8 in turn, interacting with p53, induced its stabilization and cell growth arrest [15].

In many cases the expression of TRIM8 is regulated at the post-transcriptional level by microRNA (miRNA). In clear cell Renal Cell Carcinoma (ccRCC), the miR-17-5p and miR-106b-5p targeted TRIM8 mRNA promoting its degradation [16].

Additionally, miR-182 targeted TRIM8 for degradation, and its upregulation resulted in positive or negative outcomes depending on the circumstances. In Anaplastic Thyroid Cancer (ATC), miR-182 upregulation induced TRIM8 downregulation contributing to chemoresistance, while in Airway Smooth Muscle (ASM) cells stimulated with TNFα, miR-182 upregulation attenuated the NF-κB activity and therefore the cell proliferation and migration [17,18].

Other miRNAs that targeted TRIM8 for degradation are miR-665-3p, which attenuated Oxygen-Glucose Deprivation (OGD)-induced apoptosis and inflammation in microglial cells [19] and miR-373-3p, which reduced sepsis-induced by acute hepatic injury (AHI) [20].

## 5. TRIM8 and p53 Pathway: Not Dying, but Stopping

It has been shown that TRIM8 acts as a suppressor gene in most tumors. For the first time, the link between TRIM8 and cancer was observed in glioblastoma, where it was observed that the TRIM8 genomic locus was subject to frequent deletion or loss of heterozygosity. In fact, TRIM8 was initially designated as a Glioblastoma-Expressed RING finger Protein (GERP) [21].

TRIM8 downregulation was associated with metastatic progression in Larynx Squamous Cell Carcinoma (LSCC), the most frequent neoplasm of the head and neck [22]. Moreover, TRIM8 downregulation was also found in other tumors such as osteosarcoma cell lines, ColoRectal Cancer (CRC) and Chronic Lymphocytic Leukemia (CLL) [23,24].

Later, a functional link between TRIM8 and p53 in cancer was demonstrated. Indeed, it was reported that TRIM8 is a direct p53 target gene and, in response to genotoxic stress, induces p53 stabilization and activation, leading to cell cycle arrest and reduction in cell proliferation by a positive feedback loop (Figure 2) [15]. Functionally, TRIM8 was found to physically interact with p53, preventing its interaction and degradation by Murine Double Minute 2 (MDM2), the principal negative regulator of p53. Moreover, TRIM8 was found downregulated in patients affected by chemoresistant clear cell Renal Cell Carcinoma (ccRCC). TRIM8 downregulation is due to the upregulation of miR-17-5p and miR-106b-3p, whose expression is promoted by N-MYC. Silencing of these two miRNAs restores TRIM8 expression levels, which in turn leads to p53 stabilization with the activation of cell cycle arrest and transcription of miR-34a that targets N-MYC for degradation. The final effect is the restoration of cells’ chemosensitivity to chemotherapeutic treatments [16,25,26].

Moreover, the unfavorable clinical outcome of patients affected by glioma has been correlated with TRIM8 downregulation, and restoration of TRIM8 expression reduced the clonogenic potential of the U87MG glioma cell line [27].

TRIM8 was found downregulated also in ATC tissues and cell lines due to miR-182 upregulation, which target TRIM8 contributing to chemoresistance of ATC cells [17].

Recently, TRIM8 was found downregulated in Breast Cancer (BC) and this is associated with poor prognosis. Indeed, TRIM8 knockdown significantly enhances BC cell proliferation and migration. Functionally, in the cytoplasm, TRIM8 interacts through its RING domain with AF1 domain (Activation Function 1) of Estrogen Receptor α (ERα), increasing poly-ubiquitination and inhibition of ERα [28].

TRIM8 was found differentially expressed in melanoma together with other TRIMs (TRIM2, TRIM7, TRIM18, TRIM19, TRIM27 and TRIM29), playing an important role in the development of this cancer [29].

Another important component of TRIM8′s capacity in counteracting the proliferation of cancer cells is highlighted by its effects on the stability and activity of the oncogenic transcription factor ΔNp63α, belonging to p53 gene family. ΔNp63α is upregulated in different tumors and its expression levels are correlated with a poor prognosis of patients [30,31,32]. It has been demonstrated that TRIM8 promotes ΔNp63α degradation in both proteasomal and caspase-1 dependent ways. It is important to point out that ΔNp63α is able to downregulate TRIM8 transcription expression levels, thus preventing p53 stabilization [33].

We believe that the recent discovery demonstrating a role for TRIM8 in the regulation of autophagy may be closely linked to the role of TRIM8 as a tumor suppressor, the drawback being that it also allows the survival of cancer cells due to the fact that TRIM8 has the features of a double-edged sword gene, having both oncogenic and tumor suppressor functions.

Autophagy sustains cellular fitness by eliminating dysfunctional proteins, aggregates and defective organelles, and recently it has also been demonstrated to be essential in removal of damaged DNA [34]. Following a genotoxic stress, damaged DNA is exported out of the nucleus and degraded by the lysosomes. The failure of this process leads to the activation of inflammatory pathways by intracellular DNA sensors such as cGAS/STING [35]. In this context, following genotoxic stress, TRIM8 empowers autophagy regulating lysosomal biogenesis and autophagy flux in a p53-independent manner [14]. Interestingly, TRIM8 controls the expression of p62, which has multiple functions during autophagy such as being a cargo selector, inflammation and senescence induced by DNA damage and restriction of inflammation by promoting mitophagy [14,36,37]. Functionally, during genotoxic stress, TRIM8 stabilizes XIAP (X-linked Inhibitor of Apoptosis Protein), a major regulator of cell death and autophagy, forming a trimeric complex with Caspase-3, inhibiting XIAP activation in the presence of etoposide. Interestingly, XIAP strongly activates NF-κB, which induces the expression of Beclin-1 involved in autophagy [14,38].

## 6. TRIM8, NF-κB and JAK-STAT Pathways: A Dangerous Ménage-à-Trois in Cancer

NF-κB represents a family of inducible transcription factors which regulate the expression of numerous genes involved mainly in immune and inflammatory responses. This family is composed of five structurally correlated members: NF-κB1 (alias p50), NF-κB2 (alias p52), RelA (alias p65), RelB and c-Rel, which in the form of hetero- or homodimers activate the transcription of target genes that contain the κB binding site in their regulatory regions. The NF-κB proteins are kept inhibited in the cytoplasm by a family of proteins characterized by the presence of ankyrin repeats, including the IκB family members (Figure 3). The classical activation of NF-κB is triggered by the proinflammatory cytokines Tumor Necrosis Factor α (TNFα) and InterLeukin-1β (IL-1β) and requires the phosphorylation of the NF-κB inhibitor IκBα by the activated IκB kinase complex (IKK). Phosphorylated IκBα is subject to subsequent ubiquitin proteasomal degradation. The IKK complex is composed of a regulatory subunit IKKγ/NEMO (NF–κB Essential MOdulator) and two catalytic subunits, IKKα and IKKβ. In this cascade mechanism, the role of TAK1 (TGF-β Activated Kinase 1), a serine/threonine kinase, is crucial as it phosphorylates and activates IKK, contributing to transfering the signal from the receptor to the downstream signaling molecules [39].

The proteasomal degradation of phosphorylated IκBα leads to the release of the NF-κB dimers with consequent nuclear entry (Figure 3). Once inside the nucleus, NF-κB dimers regulate genes involved in cell death inhibition and cell proliferation stimulation, thus promoting migratory and invasive phenotypes connected with tumor progression as well as Epithelial–Mesenchymal Transition (EMT).

At least two ways have been described where TRIM8 activates the NF-κB signaling pathway: one acting in the cytoplasm and the other acting in the nucleus. In the nucleus, TRIM8 promotes the translocation of PIAS3 (Protein Inhibitor of Activated STAT3) from the nucleus to the cytosol and induces its degradation. In this way, PIAS3 can no longer bind the RelA (p65) subunit of NF-κB, which is free to dimerize and activate the NF-κB responsive genes [40]. In the cytoplasm, TRIM8 empowers the activation of NF-κB triggered by TNFα and IL-1β. TNFα is the major activator of carcinogenesis and inflammatory diseases. Specifically, TRIM8 mediates the Lys63-linked polyubiquitination of TAK1 at Lys158 [41]. In turn, TAK1 activates the kinase IKK, which promotes IκBα phosphorylation and NF-κB activation [42].

TRIM8′s second partner in crime in its oncogenic role is STAT3, which belongs to the STAT family of transduction signal responsive transcription factors, which, like NF-κB, are retained in an inactive form in the cytoplasm of non-stimulated cells (Figure 3). STAT3 activation does not require the degradation of an inhibitor, as is the case for NF-κB, but is instead mediated by the phosphorylation of Tyr 705 that induces STAT3 dimerization. In this form, STAT3 enters the nucleus and promotes the transcription of different target genes. Members of the JAK family of tyrosine kinase receptors commonly mediate STAT activation, and in the case of STAT3 the major activator is JAK1. Moreover, STAT3 activity can be optimized through a reversible acetylation mechanism, which also influences the activity of NF-κB family members [43].

Both NF-κB and STAT3 are activated in response to overlapping stimuli, such as stresses and cytokines, although they are regulated by entirely different signaling mechanisms. Once activated, both NF-κB and STAT3 control the expression of pro-proliferative, immune response and anti-apoptotic genes. Some of these target genes overlap, and their transcription requires binding of both transcription factors [44]. The interaction or antagonism between NF-κB and STAT3 also occurs at the level of other signal transduction mediators, such as SOCS (Suppressor Of Cytokine Signaling) proteins, whose expression is controlled by both NF-κB and STAT3 (Figure 3). STAT3 can extend the retention of NF-κB in the nucleus, therefore SOCS-mediated STAT3 inactivation may also be responsible for NF-κB inactivation [45]. Under physiological conditions, the activation of STAT proteins is rapid and transient because they are negatively regulated by proteins such as SOCS and PIAS [46,47,48]. In turn, SOCS proteins also become unstable and seem to be rapidly degraded by proteasomal pathways. TRIM8 interacts with SOCS-1 through the SH2 domain and mediates its degradation, allowing the activation of JAK-STAT induced by IFNγ [49].

Another way in which TRIM8 activates the JAK-STAT pathway is by interacting with PIAS3 and promoting its ubiquitin proteasome degradation or exclusion from the nucleus (Figure 3) [49]. It is noteworthy that PIAS3 regulates NF-κB, as mentioned before [50], and Estrogen Receptor α (ERα) [51].

TRIM8 controls the JAK-STAT pathway also in stemness by interacting with Hsp90β, which binds STAT3 and inhibits the transcription of Homeobox protein Nanog, a transcription regulator of proliferation and self-renewal in Embryonic Stem (ES) cells. TRIM8 silencing enhances the phosphorylation of STAT3 in the nucleus leading to increased Nanog transcription [13].

The TRIM8-STAT3 pathway also regulates stemness in GSC (Glioblastoma Stem-like Cells), promoting the ubiquitination of PIAS3. It is well documented in GMB that STAT3 is a key factor that sustains the stem cell phenotype, in part through regulation of SOX2, Olig2 and Nanog, and fosters tumor cell proliferation, invasion and angiogenesis [52,53,54,55]. A TRIM8-driven transcriptomic profile in mouse neural stem cells identified TRIM8-enriched pathways which are correlated to important functions of the Central Nervous System (CNS) including the GABA receptor, axonal guidance, glutamate receptor signaling, synaptic long-term potentiation/depression and the regulatory network involving the JAK-STAT pathway [56].

## 7. The Dark Side of TRIM8

Recent studies show that TRIM8 is involved in several disorders, including Ischemia/Reperfusion (I/R) injury and cardiovascular diseases. During the reperfusion stage of ischemic tissue, the abundance of oxygen supply causes an explosion of reactive oxygen species, which induces oxidative stress as well as the formation of inflammatory mediators leading to post I/R inflammatory injury and cell death. In many cases, following this stress, the levels of TRIM8 as well as p53 increase. The activation of TRIM8 is most likely a consequence of p53 activation since we have demonstrated that TRIM8 is a p53 direct target gene [15]. However, in certain circumstances, the combined activation of TRIM8 and p53 can result in negative outcomes (Figure 2). For example, the induction of cell death following hypoxic stress caused by the absence of oxygen in a rapidly growing tumor mass is certainly a positive effect, but it is not anymore so in response to hypoxic stress due to ischemia following stroke or myocardial infarction [57]. In these cases, downregulation of both p53 and TRIM8 appears to be extremely beneficial during the early stages of ischemia or during sub-sequent reperfusion injury.

Accordingly, TRIM8 was reported to play a crucial role in regulating Oxygen-Glucose Deprivation/Re-oxygenation (OGD/R) that happens during a neuronal injury induced by insufficient cerebral blood flow. This causes irreversible brain damage that is further exacerbated by blood reperfusion (cerebral ischemia/reperfusion injury). TRIM8 expression was upregulated in neurons exposed to OGD/R, and its downregulation has a protective effect by supporting the activation of Nrf2 (Nuclear factor (erythroid-derived 2)-like 2)/ARE (Antioxidant Response Element) pathway via AMPK [58]. As further confirmation, it has been reported that p53 is activated and promotes apoptosis in OGD/R-induced injury in vitro and in cerebral ischemia/reperfusion-induced injury in vivo [58,59,60].

Moreover, it was found that TRIM8 expression increases also in microglial cells following OGD, which happens during an ischemic stroke, and in this condition the microglial cells produce pro-inflammatory cytokines causing a considerable neuroinflammatory response, leading to neuronal injury and brain damage [61]. As in other cases, the expression of TRIM8 is turned off by the action of a miRNA, which in this case is miR-665-3p. The increased expression of TRIM8 is inversely related to the expression of miR-665-3p, which attenuated OGD-induced apoptosis and inflammation in microglial cells. The downregulation of TRIM8 protected microglial cells from OGD-induced cytotoxicity and inflammation, turning off the activation of the NF-κB pathway [19].

TRIM8 expression was found greatly increased in the peri-infarct area of mice with cerebral ischemia/reperfusion injury. The suppression of TRIM8 expression, as well as of the NF-κB pathway, significantly reduces inflammation induced by ischemia, cerebral cognitive disability, apoptosis in the peri-hematoma cortex and the hippocampus. [61].

Recently, it was reported that long non-coding RNA(lncRNA) Nespas directly interacts with TAK1 to inhibit TRIM8 induced Lys63-linked polyubiquitination of TAK1 and NF-κB activation. It was observed that lncRNA Nespas significantly suppressed microglial cell death after ischemic stroke; importantly, Nespas overexpression also stopped the expression of proinflammatory cytokines that play protective roles in ischemic stroke through alleviating cell apoptosis and neuroinflammation [62].

It has also been found that TRIM8 expression in the heart was greatly upregulated in cardiomyoblast H9c2 cells after stimulation with Hypoxia/Reoxygenation (H/R), overstating cardiac hypertrophy. In H9c2 cells, TRIM8 downregulation stops the production of ROS and promotes the expression of superoxide dismutase, and glutathione peroxidase suppressed Caspase-3 activity and the expression of the pro-apoptotic Bax gene. At the same time, in H/R-stimulated H9c2 cells TRIM8 silencing causes a marked increase in Bcl-2 expression and the activation of the PI3K/Akt signaling pathway, which is well documented in being associated with a decrease in myocardial ischemic injury. PI3K/Akt activation inhibits cardiomyocyte apoptosis induced by hypoxia, and it protects against I/R injury [63].

In response to pressure overload (hypertrophic stimuli), TRIM8 translocates from the nucleus to the cytoplasm, promoting TAK1 ubiquitination and phosphorylation of IKK with subsequent cardiac hypertrophy triggered by aortic banding. The exaggerated hypertrophic response in TRIM8-transgenic hearts was accompanied by enhanced activation of p38 and JNK1/2 kinases. TRIM8 could weakly interact with TAK1 without stresses, but the interaction was significantly enhanced by Angiotensin II administration. Interestingly, in both Angiotensin II–treated cardiomyocytes and hypertrophic mouse hearts there is an increase in Interferon-γ and p53 protein levels that might be responsible for increased TRIM8 expression in hypertrophic stimulation [64].

Additionally, in mice subjected to hepatic Ischemia/Reperfusion (I/R) injury, TRIM8 expression levels increased and, conversely, TRIM8 silencing reduced hepatic inflammation and inhibited apoptosis. Functionally, TRIM8 deficiency may cause hepatic protective effects by inhibiting the activation of TAK1-p38/JNK signaling pathway [65].

In non-alcoholic steatohepatitis, TRIM8 ubiquitinates TAK1 inducing the phosphorylation and the activation of downstream c-Jun N-terminal kinase/p38/NF-κB pathways and promoting insulin resistance, hepatic steatosis and fibrosis in mouse livers [66]. Consistently, TRIM8 knockdown attenuates liver steatosis by reducing the secretion of pro-inflammatory modulators, such as IL-6, IL-1β, TNF-α and CXCL-2 with consequent inactivation of the NF-κB pathway [66,67].

Recently, it has been demonstrated that TNIP3 (TNFAIP3 Interacting Protein 3) is a new inhibitor of Non-Alcoholic-Steato-Hepatitis (NASH); TNIP3, in response to metabolic hepatic stresses, binds to TAK1 and inhibits its ubiquitination and activation mediated by TRIM8 (Figure 3) [68].

## 8. TRIM8 as Inflammation Inducer

A critical role for TRIM8 in inflammatory processes has been highlighted by proof that TRIM8 has pro-inflammatory effect in keratitis induced by *Pseudomonas aeruginosa*, a major cause of ocular morbidity that often leads to inflammatory epithelial edema with corneal ulceration and vision loss. Indeed, TRIM8 promotes Lys63-linked polyubiquitination of TAK1, leading to activation of TAK1 and of the downstream NF-κB pathway, enhancing the inflammatory responses [69].

It was found that during Acute Lung Injury induced by LipoPolySaccharide (LPS), TRIM8 expression increases in a time-dependent manner, enhancing inflammation and oxidative stress via the inactivation of p-AMPKα, while TRIM8 knockdown relieved inflammation by regulating Nrf2 signaling and Heme Oxygenase-1 (HO-1) expressions [67]. In addition, TRIM8 suppression induces down regulation of IL-1β, IL-6 and TNF-α through the inactivation of NF-κB, markedly reducing LPS-induced inflammatory response [67].

Sepsis is a systemic inflammatory response induced by severe infection that leads to a high mortality rate and a role for TRIM8 in sepsis has been reported. Long non-coding RNA LINC00472 and TRIM8 were found significantly upregulated in liver tissues and human liver THLE-3 cells in LPS sepsis-induced Acute Hepatic njury (AHI) in vitro, while miR-373-3p was downregulated. miR-373-3p targets TRIM8 for degradation, while LINC00472 acts as a sponge for miR-373-3p negatively regulating its expression levels. Downregulation of LINC00472, by modulating the miR-373-3p/TRIM8, reduced sepsis-induced AHI axis and inhibited the expression levels of main pro-inflammatory cytokines as IL-6, IL-10 and TNF-α [20].

Additionally, in OsteoArthritis (OA) chondrocytes stimulated with Interleukin 1β (IL- 1β) to induce inflammation, the expression of TRIM8 was found to be significantly increased. OA is a joint disease connected with articular cartilage degradation, subchondral bone sclerosis and osteophyte formation. Coherently, in chondrocytes stimulated with IL-1β, the knockdown of TRIM8 attenuated the production of the proinflammatory cytokines including TNF-α and IL-6 and also of the inflammatory mediators, including nitric oxide and prostaglandin E2, which are crucial inflammatory mediators and regulators of cartilage matrix synthesis and degeneration. On the other hand, the positive effects of TRIM8 silencing on IL-1β-induced chondrocytes were attributed to the inhibition of the NF-κB pathway [70].

In Airway Smooth Muscle (ASM) cells stimulated with TNFα, a model that in vitro mimics asthma, it has been shown that miR-182-5p which targets TRIM8 was markedly downregulated. By increasing miR-182-5p and accordingly decreasing TRIM8, the NF-κB activity induced by TNF-α was turned off and the proliferation and migration of ASM cells evoked by TNF-α were blocked [18].

## 9. TRIM8 Involvement in Innate Immunity

Another significant function in which TRIM8 is involved is the regulation of innate immunity.

The role of TRIM8 was investigated in commercially important farmed fish species in China and Southeast Asian countries during viral infection. The authors reported that TRIM8 overexpression inhibited fish iridovirus and nodavirus replication and was able to regulate the expression of pro-inflammatory cytokines and interferon-related signaling molecules [71].

Although a large proportion of TRIM genes were reported to be induced by Interferons (IFNs) in immune cells, few of them are activated in response to RNA virus infection in primary plasmacytoid Dendritic Cells (pDCs) playing a crucial role in the antiviral innate immune response by producing a large amount of type I interferon [72]. It has been reported that TRIM8 together with TRIM25 effectively intervene in the innate antiviral response by regulating IRF7 (IFN Regulatory Factor 7). Mechanistically, TRIM8 competes with PIN1 (Peptidyl-prolyl Isomerase 1) in binding to phosphorylated IRF7 (pIRF7), preventing its proteasomal degradation. PIN1 specifically recognizes phosphorylated Ser/Thr-Pro motifs and catalyzes the isomerization of the protein bond, influencing its function and/or the stability.

Recently, it has been reported that TRIM8 is a negative regulator of innate immune and inflammatory responses mediated by Toll-Like Receptors 3 and 4 (TLR3 and TLR4). Functionally, the authors reported that TRIM8 interacts with TRIF (Toll/IL-1 Receptor domain-containing adaptor-inducing IFN-β) mediating its Lys6- and Lys33-linked polyubiquitination, which leads to the disruption of the TRIF-TANK binding kinase-1 association [73].

## 10. TRIM8 Mutation and Diseases

Mutations in the TRIM8 gene have been associated with several rare disorders including developmental delay and intellectual disability with different degrees of severity (absent or minimal speech, delayed walking or non-ambulant and intractable epileptic seizures).

In early infantile epileptic encephalopathy, characterized by the onset of intractable epilepsy and unfavorable developmental outcomes, TRIM8 truncated variants have been identified. Some patients were reported to also have concomitant nephrotic syndrome; both these clinical manifestations are compatible with the brain and kidney expression of TRIM8 [74,75].

More recently, TRIM8 point mutations have been associated with FSGS (Focal Segmental GlomeruloSclerosis) syndrome, displaying milder neurodevelopmental problems (mild speech delay, borderline motor milestones, mild intellectual disability, small number of seizures) and more severe and progressive renal phenotype in FSGS children [76]. A single novel de novo heterozygous frameshift mutation (Tyr400Arg) [76] and another heterozygous de novo pathogenic nonsense mutation closest to the C-terminal end (C1380T>A, p.Tyr460*) [77] in the TRIM8 gene have been identified. In both cases, no significant changes in the expression of the TRIM8 protein have been observed.

These findings provide proof that truncating mutations of TRIM8 are associated with a syndrome with both neurological and renal features suggesting that patients affected by proteinuria could be therefore screened for possible TRIM8 mutations.

All reported cases of TRIM8 mutations were found to be de novo and heterozygous and were clustered in TRIM8 sixth exon, corresponding to the C-terminus of the protein. The mechanism behind the dominant feature of these mutations could be the formations of TRIM8 dimers constituted by one mutated and one wild-type monomer, in which the mutated monomer acts as dominant-negative toward the wild-type allele.

Different heterozygous putative loss of function variants in the TRIM8 gene have been listed in the Genome Aggregation Database.

## Figures and Tables

**Figure 1 cells-10-00561-f001:**
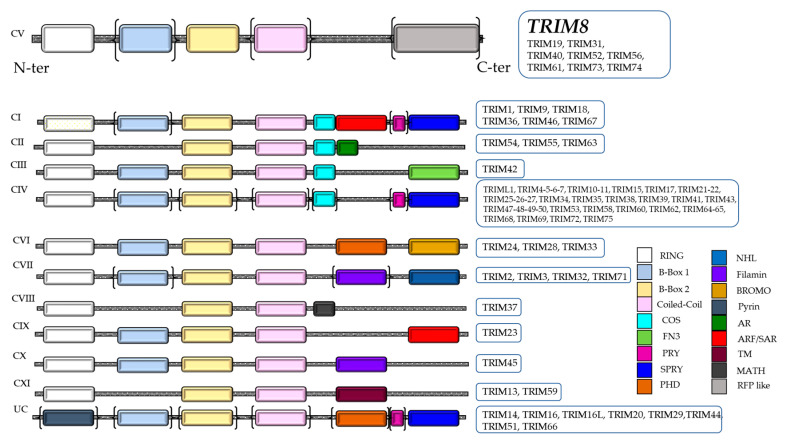
Classification of the human TRIpartite Motif-containing (TRIM) protein family, composed of 11 sub-families, from C-I to C-XI. The UC (UnClassified) subfamily indicates TRIM proteins unclassified for the absence of the Really Interesting New Gene (RING) finger domain. Additional domains are NHL: NHL repeats; COS: COS box motif; FN3: fibronectin type III motif; PHD: plant homeodomain; BROMO: bromodomain; MATH: meprin and TNF-receptor associated factor TRAF homology domain; TM: transmembrane domain; AR: acid-rich region; RFP-like domain. The domains in brackets indicate that they can be absent.

**Figure 2 cells-10-00561-f002:**
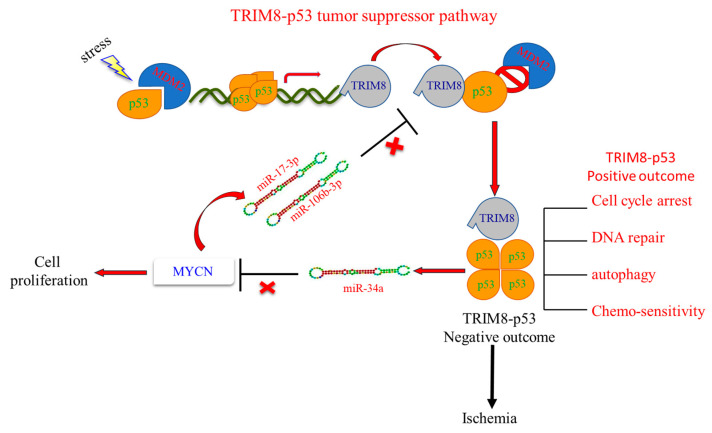
The TRIM8-p53 pathway. Following genotoxic stress, p53 promotes TRIM8 transcription and, through a positive feedback mechanism, TRIM8 stabilizes and activates p53, inducing the expression of genes involved in cell cycle arrest and DNA repair. Furthermore, p53 promotes the transcription of miR-34a, which in turn inhibits the activity of MYCN and the expression of miR-17-3p and miR-106b-3p. Therefore, TRIM8 is no longer silenced by these miRNAs. The final effect is the p53-mediated cell cycle arrest and the restoration of cells’ chemosensitivity to therapeutic treatments. However, in certain circumstances, the combined activation of TRIM8 and p53 can result in negative outcomes as in response to hypoxic stress due to ischemia following stroke or myocardial infarction.

**Figure 3 cells-10-00561-f003:**
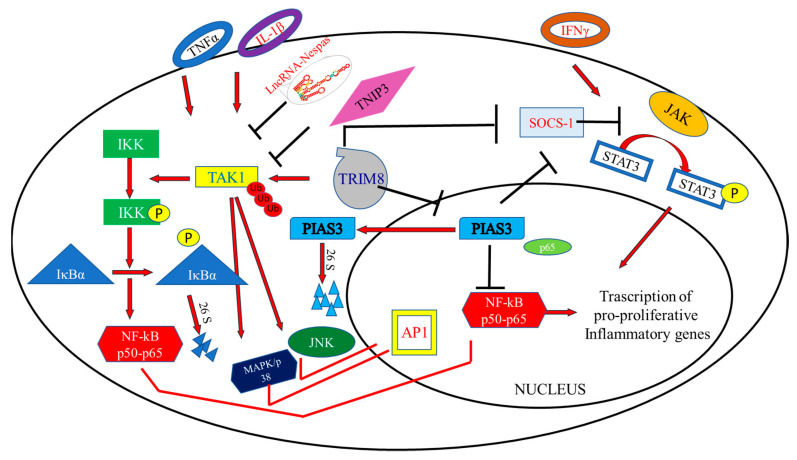
The TRIM8, NF-κB and JAK-STAT pathways. Pro-inflammatory cytokines (TNFα e IL-1β) promote NF-κB activation through TRIM8. In the nucleus, TRIM8 promotes the translocation of PIAS3 in the cytoplasm where it is then degraded. PIAS3 in the nucleus interacts with RelA (alias p65) preventing NF-κB activation. Moreover, TRIM8 promotes TAK1 Lys63- linked polyubiquitination leading to IKK kinase activation. IκBα phosphorylation mediated by IKK leads to its degradation and NF-κB translocation in the nucleus. TAK1 also induces the activating phosphorylation of downstream MAPK/p38 and JNK signaling pathways. Long non-coding RNA Nespas directly interacts with TAK1 and inhibits TRIM8-induced Lys63- linked polyubiquitination. Furthermore, TRIM8 induces the activation of the JAK-STAT pathway promoted by IFN-γ through the degradation of two STAT protein inhibitors, PIAS3 and SOCS-1.

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
