# Peer review of "Emerging Roles of TRIM8 in Health and Disease"

_cells, 2021, doi:10.3390/cells10030561_

Round 1
Reviewer 1 Report
The review by Marzano et al represents a comprehensive overview of the various roles of TRIM-8 in health and disease. The content was interesting and contained a lot of information. However, I feel that some restructuring in parts may help the non-specialist reader follow the subject area. For example, throughout, regulation of TRIM-8 by microRNAs was discussed. This could possibly be brought together in a section on TRIM-8 regulation at the transcriptional and post-transcriptional level. Bringing some aspects together on TRIM-8 targets and their role in disease may also help.
Some specific points are outlined below:
- An introductory summary of the review, and the salient new information discussed, would be useful before the section on ubiquitination.
- It would be useful to have a diagram of protein domains within TRIM family proteins, with TRIM-8 highlighted.
- Figure 2-clearer would be clearer if the TRIM-mediated polyubiqitination of TAK-1 was indicated on the figure.
- It would help make the review clearer and more succinct if the role of the protein in inflammation, innate immunity etc was combined with its role in various diseases. Keeping them separate resulted in overlap.
Author Response
Dear Editor,
Thank you for considering our manuscript (Manuscript ID: cells-1102381): “Emerging roles of TRIM8 in health and disease” by Flaviana Marzano et al. for publication in Cells-Special Issue " "Cellular Function of TRIM E3 Ubiquitin Ligases in Health and Disease"”.
We sincerely thank you and the reviewers for constructive criticism and valuable comments, which were of great help in revising the manuscript.
We have carefully considered the reviewer's comments and address their points to fulfil their requests.
The revised version of the manuscript contains all the revisions required with a new figure.
All the revisions in the manuscript are in track changes mode.
Your Sincerely,
Apollonia Tullo
REVIEWER 1
We really thank the reviewer for the insightful comments and great suggestions. We have carefully evaluated them to address the points raised and fulfil the requests.
All the revisions in the manuscript are in track changes mode.
Below our answers and clarifications point-by-point.
- The review by Marzano et al represents a comprehensive overview of the various roles of TRIM-8 in health and disease. The content was interesting and contained a lot of information. However, I feel that some restructuring in parts may help the non-specialist reader follow the subject area. For example, throughout, regulation of TRIM-8 by microRNAs was discussed. This could possibly be brought together in a section on TRIM-8 regulation at the transcriptional and post-transcriptional level. Bringing some aspects together on TRIM-8 targets and their role in disease may also help.
- Answer
We thank the reviewer for the positive comments and valuable suggestions on our review. As suggested by the reviewer, we have included a new paragraph on Regulation of TRIM8 gene expression.
2.“An introductory summary of the review, and the salient new information discussed, would be useful before the section on ubiquitination”.
Answer
- We thank the reviewer for this comment. We modified the abstract by inserting an introductory summary with salient new information discussed.
3.“It would be useful to have a diagram of protein domains within TRIM family proteins, with TRIM-8 highlighted”.
Answer
- We thank the reviewer for this suggestion. We share the idea that a diagram with protein domains can help the reader. Therefore, we inserted a new Figure (Figure 1) showing the different domains of the TRIM proteins with TRIM8 highlighted.
4.“Figure 2-clearer would be clearer if the TRIM-mediated polyubiqitination of TAK-1 was indicated on the figure”.
Answer
- We modified the Figure 2, indicating the polyubiqitination of TAK-1 mediated by TRIM8.
5.“It would help make the review clearer and more succinct if the role of the protein in inflammation, innate immunity etc was combined with its role in various diseases. Keeping them separate resulted in overlap”.
Answer
- We thank the reviewer for this comment, but we believe it is better to leave them separate because
in the paragraph "The dark side of TRIM8" we reported those pathologies such as Ischemia / Reperfusion (I / R) injury and cardiovascular diseases in which the levels of TRIM8 as well as p53 increased, discussing that in certain circumstances, the combined activation of TRIM8 and p53 can result in negative outcomes. For example, the induction of cell death following a hypoxic stress caused by the absence of oxygen in a rapidly growing tumor mass, is certainly a positive effect but it is not anymore so in response to hypoxic stress due to ischemia following stroke or myocardial infarction. In these cases, downregulation of both p53 and TRIM8 appears to be extremely beneficial during the early stages of ischemia or during subsequent reperfusion injury.
Reviewer 2 Report
The manuscript of Marzano et al presents a comprehensive, detailed, in-depth review of TRIM8 protein, a member of the large family of TRIM E3 ubiquitin ligases. The authors thoroughly describe the signaling mechanisms (p53, NFkB and JAK/STAT pathways), biological processes (survival/apoptosis, differentiation, proliferation, autophagy, innate immunity) and pathological mechanisms (cancer, ischemia/reperfusion, inflammatory diseases) in which TRIM8 is implicated. I find this review well-written and useful for readers working in this field and for those as well who just want to get insight into this important aspect of cellular regulation.
Just a few critical comments:
- The fact that TRIM proteins are E3 ub ligases should be mentioned in the Abstract.
- What the abbreviation TRIM stands for (Tripartite motif) should be given in the main text where it first appears (line 78).
- It would be helpful for the reader to have a simple figure on the domain structure of the subfamilies of TRIM proteins (preferably in Chapter 2).
- Abbreviations should be explained once, where they first appear. Some of them are given several times (e.g. PIAS: lines 248, 277, 282, TAK1: lines 226, 338, TNF: lines 222, 407). (A List of abbreviations as a footnote would be useful.)
- Some abbreviations appear only once (e.g. Pseudomonas aeruginosa (PA), line 379, acute lung injury (ALI), line 383, etc), they should be omitted.
- Typing/grammatic errors that should be corrected: lines 35/36:...about 3% of human protein..., line 67: It is interesting to note..., line 112: ...domain, a Coiled-Coil..., lines 217/218:...homodimers activate the transcription..., line 259: ...as is the case for..., line 261:...STAT3 enters the nucleus, line 390/391:...Long non-coding RNA LINC00472..., Line 394:...LINC00472 acts as a sponge for miR-373-3p..., line 440:...characterized by the onset of intractable...
Author Response
Dear Editor,
Thank you for considering our manuscript (Manuscript ID: cells-1102381): “Emerging roles of TRIM8 in health and disease” by Flaviana Marzano et al. for publication in Cells-Special Issue " "Cellular Function of TRIM E3 Ubiquitin Ligases in Health and Disease"”.
We sincerely thank you and the reviewers for constructive criticism and valuable comments, which were of great help in revising the manuscript.
We have carefully considered the reviewer's comments and address their points to fulfil their requests.
The revised version of the manuscript contains all the revisions required with a new figure.
All the revisions in the manuscript are in track changes mode.
Your Sincerely,
Apollonia Tullo
REVIEWER 2
We really thank the reviewer for the insightful comments and great suggestions. We have carefully evaluated them to address the points raised and fulfill the requests.
All the revisions in the manuscript are in track changes mode.
1.The fact that TRIM proteins are E3 ub ligases should be mentioned in the Abstract.
1.Answer
We agree with the reviewer and mentioned that TRIM proteins are E3 ub ligases in the Abstract.
- What the abbreviation TRIM stands for (Tripartite motif) should be given in the main text where it first appears (line 78).
2.Answer
We thank the reviewer for pointing out this oversight. We indicated in the text that the abbreviation TRIM stands for (Tripartite motif) where it first appears.
- It would be helpful for the reader to have a simple figure on the domain structure of the subfamilies of TRIM proteins (preferably in Chapter 2).
3.Answer
We thank the reviewer for this suggestion. We share the idea that a diagram with protein domains can help the reader. Therefore, we inserted a new Figure (Figure 1) showing the different domains of the TRIM proteins with TRIM8 highlighted.
- Abbreviations should be explained once, where they first appear. Some of them are given several times (e.g. PIAS: lines 248, 277, 282, TAK1: lines 226, 338, TNF: lines 222, 407). (A List of abbreviations as a footnote would be useful.)
4.Answer
We thank the reviewer and apologize for these oversights. We amended the text as indicated and we added a list of abbreviations as a footnote.
5.Some abbreviations appear only once (e.g. Pseudomonas aeruginosa (PA), line 379, acute lung injury (ALI), line 383, etc), they should be omitted.
5.Answer
We thank the reviewer for pointing out this oversight. We deleted the abbreviation that appear only once such as Pseudomonas aeruginosa (PA), line 379, acute lung injury (ALI), line 383.
6.Typing/grammatic errors that should be corrected: lines 35/36:...about 3% of human protein..., line 67: It is interesting to note..., line 112: ...domain, a Coiled-Coil..., lines 217/218:...homodimers activate the transcription..., line 259: ...as is the case for..., line 261:...STAT3 enters the nucleus, line 390/391:...Long non-coding RNA LINC00472..., Line 394:...LINC00472 acts as a sponge for miR-373-3p..., line 440:...characterized by the onset of intractable...
6.Answer
We really thank the reviewer and apologize for these oversights. We amended the text and check all the typing/grammatic errors.